# Prevalence and risk factors of cutaneous leishmaniasis in a newly identified endemic site in South-Ethiopia

Behailu Merdekios[1]*, Mesfin Kote[1], Myrthe Pareyn[2], Jean-Pierre Van Geertruyden[3], Johan van Griensven[2]

1 Arba Minch University, Arba Minch, Ethiopia, 2 Institute of Tropical Medicine, Antwerp, Belgium, 3 University of Antwerp, Antwerp, Belgium

* merdekib@tcd.ie

## Abstract

**Data Availability Statement:** All relevant data are within the manuscript and its  Supporting Information files.

### Background

Although there are several areas in southern Ethiopia environmentally favourable for cutaneous leishmaniasis (CL), studies on the existence and risk factors of CL are lacking beyond a few well-known hotspots. This study aimed to assess the prevalence and risk factors of CL in Bilala Shaye, a village in the southern Ethiopian highlands at an altitude of 2,250 meters.

### Methods

A cross-sectional house-to-house survey was done between July-August 2021. Those with skin lesions were clinically assessed and data on individual risk behaviour and environmental and household features were collected using questionnaires. Univariate and multivariable logistic regression models were used to identify independent risk factors of CL at a 5% significance level with two-sided P-values <0.05 considered statistically significant.

### Result

A total of 1012 individuals were interviewed; the median age was 23 years (interquartile range 12–50), with 7% below the age of five; 51% were female. All households had domestic animals, and for 143 (57%) households goats/sheep lived inside or around the house. Animal dung was found in the compounds of 194 (77%) households. The overall prevalence of active CL was 2.5% (95% confidence interval (CI) 1.6–3.6), reaching 6.7% (95% CI 3.6–11.2) in children between 5–12 years old. The prevalence of CL scars was 38.5% (95% CI 35.5–41.6). In multivariate analysis, the presence of animal dung in the compound (adjusted odds ratio (OR) 2.1; 95% CI: 1.3–3.5, P = 0.003) and time spent outside in the late evening in areas where hyraxes live (adjusted OR 2.4; 95% CI: 1.7–3.3, P <0.001) were identified as independent risk factors.

**Funding:** BM was beneficiary of the financial grant from Vlir-uos through the Inter University Cooperation (IUC) agreement with Arba Minch University (AMU ET2017IUC035A101). URL: https://www.vliruos.be/en/project_funding/4 The funders had no role in study design, data collection and analysis, decision to publish, or preparation of the manuscript.

**Competing interests:** The authors have declared that no competing interests exist.

## Conclusion

This is the first report on the existence of CL in this village, with the high prevalence of CL scars indicating long-term endemicity. Further studies are needed to understand the role of animals and their dung in (peri)-domestic CL transmission.

## Introduction

Cutaneous leishmaniasis (CL) is a skin disease caused by *Leishmania* parasites and transmitted by the bite of female phlebotomine sandflies. CL can be a disfiguring and stigmatizing condition, especially for extensive and/or facial lesions [1]. The disease has a significant impact on the quality of life and socio-economic status of affected individuals and communities [2]. According to the latest report of the World Health Organization (WHO), 98 countries are endemic for CL and more than a million new cases of CL occur annually. Around 70–75% of the global burden of CL can be found in four countries of the New World and six of the Old World, including Ethiopia. In the Old World, *Leishmania (L.) major* and *L. tropica* are the most common species causing CL.

In Ethiopia, CL is a relevant public health problem, with an estimated annual caseload of 20,000 to 50,000 [2] although this may be an underestimation [3, 4]. Unlike most other countries in the Old World, CL in Ethiopia is predominantly caused by *Leishmania aethiopica*, and tends to be relatively more clinically diverse, severe and difficult to treat [5]. CL mainly occurs in the highlands bordering the Ethiopian Great Rift Valley, where hyraxes are thought to be the reservoir, and *Phlebotomus longipes* and *P. pedifer* the predominant vectors [6]. The epidemiology of CL in Ethiopia remains poorly understood and only few studies have looked in detail into CL risk factors. Several studies identified proximity to areas where hyraxes reside, such as gorges, as a risk factor for CL [7, 8]. Studies also hinted at the increased risk of CL by keeping domestic animals around the house, but did not analyse by type of domestic animals or adjust for other factors in the analysis [9, 10]. Other risk factors included the presence of cracks in the walls of the house, the presence of animal dung near the household and activities outside the house at night. All these factors are thought to increase the chance of being exposed to infected sandflies in or around the house, or of spending time close to hyrax areas at night when sandflies are most active [11].

Several well-defined CL hotspots have been documented in the north and the south of the country [5, 9, 10, 12–14]. Ochollo village is one of the best-studied foci in southern Ethiopia [11, 13, 15]. Over the last decades, repeated surveys were carried out, demonstrating that the caseload remains high, with up to 70% of inhabitants having suffered from the disease [13]. There are however several areas in southwestern Ethiopia with similar environmental and geographical features from which CL is not reported [9, 14]. We recently surveyed the wider region and found the vector and CL cases in 38 villages previously not reported to be CL endemic. This included Bilala Shaye, a village close to Ochollo but separated by a steep valley [16]. To further investigate this, we conducted a household survey in this village to determine the prevalence and risk factors of CL. This research aimed to contribute to a better understanding of the epidemiology of CL in Ethiopia and inform public health interventions to control the disease.

## Methods

### Study area

The study was conducted in the rural village of Bilala-Shaye, located 53 km away from Arba Minch city in the Gamo administrative zone of southwestern Ethiopia (Fig 1). Bilala-Shaye has

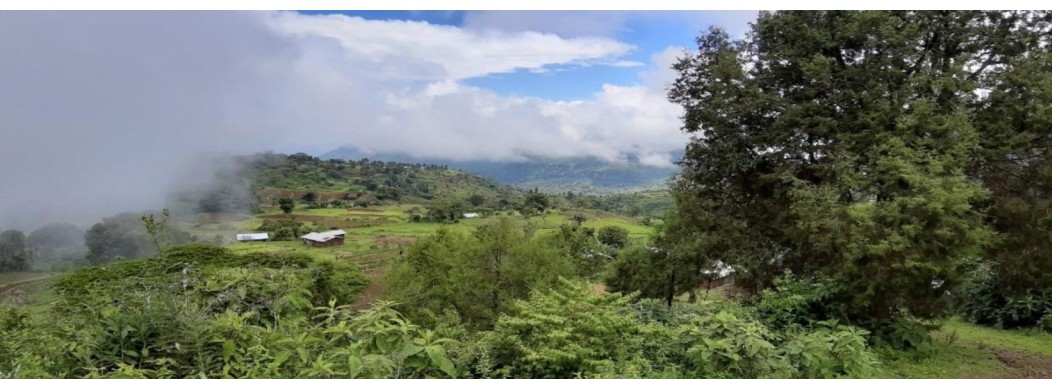

**Fig 1. Picture showing the study site, Bilala Shaye, Ethiopia 2021.**

252 households and a population of 1012. The village is situated at an average altitude of 2200 meters above sea level, ranging between 1982–2608 meters. The study area covers approximately 13 square kilometres and is divided into two sub-villages, Bilala and Shaye. The landscape of the village is hilly on the Bilala sub-village side, while the Shaye sub-village has a central plateau and rocky cliffs where hyraxes reside on the southwest, south, and east borders. Most families are involved in farming and keep livestock. The village has no electric power supply or sewerage system. Water supply is by a communal distribution point in the village. The village has one health post run by health extension workers.

## Study design

A cross-sectional survey was conducted between July and August 2021 including all households in the village.

## Conduct of the survey

The community was informed about the study through face-to-face discussions with the village administration, which was supported by an official collaboration letter from Arba Minch University (AMU). In order to maximize the number of participants to be found at home during the survey, market days were skipped. The survey was conducted by experienced data collectors from the AMU Health and Demography Surveillance research unit (HDSS). They were additionally trained by the principal investigator on how to conduct a physical assessment of individuals with skin lesions and how to clinically recognize CL, differentiating it from other skin diseases using practical exercises in the clinic, videos and hand-held pictures, which were also available and used during the survey reference. A semi-structured questionnaire was designed and uploaded to a tablet with Open Data Kit software for data collection.

In each household, all family members were examined by the study team for the presence of active lesions compatible with active CL or healed CL (scars). The following information was gathered at the household level: house construction features (wall, roof, floor), presence of cracks in the walls, a kitchen inside the house (which is often related to smoke), number of rooms and people in the same house, environmental factors such as the distance in meters between the location of the house and nearby farmland, presence of gorges or areas where hyraxes reside (within 300 meters radius), as described before [9], production of smoke at night inside the house, the presence of bed-nets, and features of the household compound (presence of animal dung or burrows, living together with domestic animals, stony fence, latrine availability).

At the individual level, the following information was gathered: sociodemographic characteristics (age, sex, educational level, marital status, and religion) adapted from the Demography and Health Survey [17]; activities outside the house after dark, particularly in areas where hyraxes reside; and use of bed nets. For clinically diagnosed CL lesions or scars, the location, number, clinical features and lesion duration were noted.

All individuals clinically diagnosed as CL with active lesions were invited to return later at the sample collection site for confirmatory PCR testing, but only 9/25 presented and underwent further sampling using non-invasive D-squame tape stripping discs (Monaderm, Monaco). The sample was collected by an experienced laboratory technician by pressing the tape on the border of the lesion for 10 seconds and carefully peeling it off [18, 19]. Samples were stored at -40˚C until further analysis.

## Laboratory analysis

Molecular analyses were performed at the Leishmaniasis Research and Treatment Centre, University of Gondar Hospital, Ethiopia. Tape discs were extracted using the Maxwell 16 LEV Blood DNA kit (Promega, Leiden, The Netherlands). The kit's lysis buffer was added to the tape disc followed by proteinase K, after which the sample was vigorously vortexed and incubated at 56˚C at 400 rpm for 20 minutes. The mixture was then loaded into the Maxwell 16 device (Promega) according to the manufacturer's instructions. In the extraction batch, a negative extraction control (lysis buffer only) was included. The samples were tested for *Leishmania* parasites by PCR targeting kinetoplast DNA (kDNA) as described before [20] on the RotorGene Q instrument (Qiagen, Venlo, The Netherlands). Positive and negative PCR controls and the negative extraction controls were included in the PCR run and results were expressed in cycle threshold (Ct) values. All negative samples were subjected to a PCR targeting the human beta-globin gene as described before [21] to evaluate the extraction efficiency and PCR inhibition.

## Statistical analysis

EpiData (version 2.2.3.187, EpiData Association, Odense, Denmark) was used for data entry of the variables mentioned above under 'conduct of the study' and Stata 15 for analysis. Data were summarized using frequencies and proportions for binary/categorical variables and medians (interquartile ranges (IQR) for continuous variables. As the number of active CL cases was low, a composite measure of active and past CL was taken as an outcome in the main analysis. Factors associated with the outcome were determined using mixed-effects logistic regression. Factors with a P-value < 0.05 in univariate analysis were included in multivariate analysis; with a back-ward selection process we maintained factors with P-values < 0.05. We calculated variance inflation factors and assessed biologically plausible interactions. To take into account, the clustering of the individuals at the household level, a mixed effects model was constructed, with a random intercept for households. In exploratory analysis, the variable 'spending time outside in the late evening in areas where hyraxes reside' was replaced by the individual late evening outdoor activities, to explore which activities were associated with an increased risk of CL.

We used both active CL and CL scars as outcomes in the main analysis, as 1) most other studies in Ethiopia have combined CL lesions and scars as outcomes, and by following this, our findings could be compared with the findings from these studies; 2) this increased the number of events to a sufficiently high number to conduct robust statistical analysis. We note that the presence of CL scars also indicates "exposure" to "CL risk factors", albeit more distant in time compared to active CL. To assess the robustness of our findings, and minimize the

effect of this "exposure distant in time", we conducted a sensitivity analysis including all active CL lesions (adults and children), and all scars in children < 12 years old. As most active CL seemed to occur in the age group 5–12 years, this would indicate "exposure to CL risk factors" not too long ago.

### Ethical issues

The study was approved by the Institutional Ethical Review Board (IRB) of the College of Medicine and Health Sciences of Arba Minch University Committee (IRB of CMHS/AMU Ref no. 11679/111). Verbal consent was obtained from all adults or parents/guardians of young children who could not read and write; written consent was obtained from literate adults. The study population was officially informed about the type and purpose of the study through its village administration. The study team ensured that participants understood the study's objectives, procedures, potential risks, and benefits before obtaining their consent. The team also emphasized that participation was voluntary, and participants had the right to withdraw from the study at any time without any consequences.

## Results

### Individual and household characteristics

A total of 1012 individuals from 252 households were included, with a 100% response rate. The median age was 23 years (IQR 12–50), with 67 (7%) below the age of five and 193 (19%) between 5–12 years old; 51% were female. About a quarter (24%) were farmers, 305 (30%) were school-going and 91 (9%) were children below school age. Overall, 42% were illiterate (Table 1).

The median household size was 3.5 members (IQR 2–6). Latrines were available in 85% of the households and 29% had a kitchen within the house (S1 Table). In 86% of the households, smoke was produced in the sleeping area in the evening or night. More than half of the houses had an iron sheet roof (65%), walls made from wood, grass, and mud (61%), and all had a floor made from soil, mud, and dung. Ownership of domestic or livestock animals was common, with 57% of the households having goats/sheep and 44% having chickens within the house or on the compound. Animal dung was found near the house or on the compound of 77% of the households. 58% of the houses were located within 300 meters of areas where hyraxes live and 44% within 300 meters from a gorge (S1 Table).

Around 47% of the participants spent time outside the house in the (late) evening (S1 Table). Evening activities included fetching water/firewood (n = 182; 18%), farming (n = 161; 16%), and herding animals (n = 98, 10%). Such activities were rare below the age of five but clearly increased after that (S2 Table). Whereas females were more likely to engage in the collection of water/firewood in the evening, males were more commonly involved in farming and herding animals. The proportion of inhabitants spending time outside in the evening in areas where hyraxes reside similarly increased from the age of 5 years onwards. All four types of late evening outdoor activities were strongly associated with spending the evening time in areas where hyraxes reside (S3 Table). Sleeping under a bed net at night was only done by 27 (2.7%) inhabitants.

### Prevalence of CL

We identified 25 active CL cases, with a median age of ten years (interquartile range 6–37), yielding a prevalence of active disease of 2.5% (95% confidence interval (CI) 1.6–3.6). Active CL was most common in children between 5–12 years old, occurring in 6.7% (95% CI 3.6–

**Table 1. Socio-demographic characteristics of the individuals included in the cutaneous leishmaniasis prevalence survey, Bilala Shay, Ethiopia 2021 (N = 1012).**

| Variable | n (%) or median (IQR) |
|---|---|
| **Age (years), median (IQR)** | 23 (12–50) |
| < 5 | 67 (7) |
| 5–12 | 193 (19) |
| 13–17 | 136 (13) |
| 18–29 | 155 (15) |
| 30–45 | 146 (14) |
| 45–65 | 198 (19) |
| ≥ 65 | 117 (12) |
| **Sex** | |
| Male | 499 (49) |
| Female | 513 (51) |
| **Marital status** | |
| Single | 280 (36) |
| Married | 382 (50) |
| Widowed/separated | 105 (14) |
| Not applicable (<18 years) | 245 |
| **Religion** | |
| Protestant | 769 (76) |
| Orthodox | 243 (24) |
| **Education** | |
| Illiterate | 423 (42) |
| Primary education | 255 (25) |
| Junior secondary education | 95 (9) |
| Secondary/higher education | 148 (15) |
| Not applicable (< 6 years) | 91 (9) |
| **Occupation** | |
| Farmer | 247 (24) |
| House-wife | 204 (20) |
| Studying | 305 (30) |
| Weaver | 51 (5) |
| Daily labourer | 17 (2) |
| Children below school age | 91 (9) |
| Unemployed/Other | 97 (10) |

IQR: interquartile range

11.2). Lesions were predominantly located on the face, with 40% on the cheek, 20% on the forehead, and 12% on the nose. The lesion duration at the time of the study was 1.8 months (IQR 0.9–2.7). In eight out of the nine active CL cases undergoing sample collection, PCR was positive for *Leishmania*. Overall, 390 cases with CL scars were found, with a median age of 32 years (IQR 16–45), giving a prevalence of previous CL of 38.5% (95% CI 35.5–41.6) (Table 2).

## Risk factors for CL

Based on the univariate analysis, a statistically significant association with CL was found with outdoor activities in the evening, increasing age, keeping goats or sheep in or around the house, the presence of animal dung around the house, and spending time outside in the

**Table 2. Localization of lesions and prevalence of cutaneous leishmaniasis (active and scar) per age group, sex, and location of lesion and scar in the individuals included in the prevalence survey, Ethiopia 2021.**

| Age (years) | Total; n = 1012 (column %) | Active CL; n = 25 (row %) | CL scar; n = 390 (row %)* |
|---|---|---|---|
| < 5 | 67 (7) | 1 (1.5) | 5 (7.5) |
| 5–12 | 193 (19) | 13 (6.7) | 65 (28.5) |
| 13–17 | 136 (13) | 2 (1.5) | 60 (44.1) |
| 18–29 | 155 (15) | 2 (1.3) | 70 (45.2) |
| 30–44 | 146 (14) | 3 (2.0) | 58 (39.7) |
| 45–64 | 198 (19) | 2 (1.0) | 83 (41.9) |
| ≥ 65 | 117 (12) | 2 (1.7) | 59 (50.4) |
| **Sex** | | | |
| Male | 499 (49.3) | 16 (3.2) | 183 (36.7) |
| Female | 513 (50.7) | 9 (1.8) | 207 (40.3) |
| **Localisation** | | | |
| Forehead | 5 | 5 (20) | |
| Cheek | 10 | 10 (40) | |
| Ears | 1 | 1 (4) | |
| Chin | 2 | 2 (8) | |
| Eye/Eyelids | 2 | 2 (8) | |
| Nose | 3 | 3 (12) | |
| Hands and Arms | 1 | 1 (4) | |
| Legs | 1 | 1 (4) | |
| Abdomen | 0 | 0 (0) | |

CL: cutaneous leishmaniasis

* there were 382 participants with only a CL scar and 8 with both a CL scar and active CL.

evening in areas where hyraxes reside (S4 and S5 Tables). Outdoor activities during the evening associated with an increased risk of CL included playing, fetching water/firewood, herding animals, and farming. Other factors such as having a house with an iron sheet roof, a kitchen inside the main house and the presence of a stone fence around the house were also associated with an increased risk but did not reach statistical significance (S4 Table).

In multivariate analysis, spending time outside the home in the evening at places where hyraxes reside (adjusted OR 2.4, 95% CI: 1.7–3.3; P-value <0.001), and the presence of animal dung on the compound (adjusted OR 2.1, 95% CI: 1.3–3.5; P-value 0.003) were associated with increased CL risk. Outdoor activities during the evening that were associated with CL included farming, collecting water/firewood, and herding animals (Table 3).

In the sensitivity analysis we used active CL lesions (adults and children) and scars in children ≤ 12 years of age as the outcome. This reduced the number of events from 407 to 80. For both the presence of dung and spending time in the evening in areas where hyraxes reside, the strength of the association increased slightly, but P-values were higher, remaining statistically significant only for the latter (Table 3).

## Discussion

We report on the burden of CL in a south Ethiopian village previously not identified to be CL endemic [16]. Active CL was found in 2.5% of the inhabitants, most commonly in the 5–12 years old. CL scars were found in 39%, predominantly in adults. Spending time outside in the evening (close to where hyraxes reside) and the presence of animal dung around the house were identified as independent risk factors for CL.

**Table 3. Independent risk factors for cutaneous leishmaniasis (CL) in the CL prevalence survey, Bilala Shaye, Ethiopia 2021 (N = 1012).**

| Risk factors | Main analysis | | | | Sensitivity analysis | | | |
|---|---|---|---|---|---|---|---|---|
| | Active CL or scar | | | | Active CL, or scar in children < 12 years | | | |
| | aOR (95% CI) | P-value | aOR (95% CI) | P-value | aOR (95% CI) | P-value | aOR (95% CI) | P-value |
| **Animal dung around the house** | | | | | | | | |
| No | 1 | **0.003** | 1 | **0.006** | 1 | 0.060 | 1 | 0.058 |
| Yes | 2.1 (1.3–3.5) | | 2.0 (1.2–3.3) | | 3.0 (0.9–9.2) | | 3.1 (1.0–9.9) | |
| **Time spent outside late evening in the area where hyraxes live** | | | | | | | | |
| No | 1 | <**0.001** | - | | 1 | **0.001** | - | |
| Yes | 2.4 (1.7–3.3) | | - | | 3.1 (1.6–5.8) | | - | |
| **Activities outside late evening** | | | | | | | | |
| Not going out | - | | 1 | <**0.001** | - | | 1 | <**0.001** |
| Playing | - | | 0.5 (0.2–1.4) | | - | | 0.6 (0.1–3.4) | |
| Fetching water/ firewood | - | | 2.2 (1.4–3.4) | | - | | 5.0 (2.3–10.9) | |
| Herding animals | - | | 1.8 (1.1–3.1) | | - | | 2.3 1.0–5.6) | |
| Farming work | - | | 2.1 (1.3–3.3) | | - | | 1.3 (0.3–5.3) | |

CL: cutaneous leishmaniasis; aOR: adjusted odds ratio; CI: confidence interval.

* The following variables (with a P-value <0.05 in univariate analysis) were included in the multivariate model: the presence of animal dung on the compound, spending time outside in the evening in areas where hyraxes reside, individual age, spending time outside during late evening and the presence of goats/sheep on the compound; via back-ward selection, variables with a P-value > 0.05 in the multivariate model besides age were removed, with only the variables with a P-value < 0.05 retained in the final model and shown in this Table. Analysis was adjusted for clustering at the household (rho 0.17 (95% CI 9.6–27.6). In exploratory analysis, the final model was rerun replacing the variable 'spending time outside late evening in areas where hyraxes live' with the individual late evening outdoor activities, to explore which activities were associated with an increased risk of CL. In sensitivity analysis, all active lesions (children and adults) and scars in children below the age of 12 years were taken as the outcome.

Our prevalence of active and past CL is comparable to some other known CL endemic places with active transmission but different from other studies. Two surveys were conducted in Tigray in the North of Ethiopia and they found active CL in 2.3% and 6.7% of the study population; CL scars were present in 21% and 7% [9, 12]. As to the South of the country, studies are restricted to Ochollo village—a different village than our study site and a well-known CL hotspot not very far from our study village. A household survey conducted 20 years ago found a prevalence of active CL of around 4% and scars in 34% of the inhabitants [22]. The most recent study focussing on primary school children in the village found active lesions in 4% and scars in 60% [13].

While the detection of active CL lesions indicates ongoing transmission, the high prevalence of CL scars including in older individuals suggests that CL has been present in the village and surroundings for decades. In a rapid survey, we previously identified a fairly high number of villages in an area in the South of Ethiopia that, based on geographical and environmental/ meteorological features (altitude and rainfall), were considered at high risk of CL in a mapping study [16]. While there are likely many longstanding CL hotspots not yet identified, there are also indications that CL is spreading. Repeated CL outbreaks have been identified over the last years, all incrementing *L aethiopica* [10, 23]. This indicates that inhabitants from many at-risk areas currently not formally considered CL endemic are likely to suffer from CL. Our findings call for better national surveillance systems and national evaluations in CL risk areas to define the current epidemiology and the need for care in the country.

In line with previous studies, we found that spending time in the late evening in areas where hyraxes reside was associated with CL. Several other risk factors probably represent a

proxy of this, such as the presence of gorges or caves close to home. These have been found associated with CL in some studies, but not in others including ours [9, 24, 25].

Keeping livestock was a risk factor in several studies outside Ethiopia and was found to be associated with CL in univariate analysis in two studies in Ethiopia [10, 25]. In our study, keeping goats in or around the house was associated with CL in univariate analysis. This could be via the attraction of sandflies or the production of dung–identified as a risk factor for CL in our and other studies. Why this association was only observed with goats/sheep is also not clear. Whether goats or other (domestic) animals besides hyraxes play a role as reservoir can currently not be fully ruled out and requires further work. We note that though the species was *L. tropica*, CL has been diagnosed in a goat in Kenya [26] and in a reservoir study in Ochollo, we previously detected *Leishmania* parasites at the nose of one single goat [27].

The presence of dung within the compound was linked to a higher risk of CL. It is yet to be determined whether this heightened risk is due to creating favourable breeding sites for sandflies, indicative of inadequate environmental sanitation conditions, or simply a result of keeping (more) animals in proximity to the living areas. Further investigations are required to confirm these potential associations. More in-depth epidemiological studies specifically looking at the association between CL, keeping animals, and the presence of animal dung is warranted, combined with studies on the breeding and feeding behaviour of (infectious) sandflies. Additionally, in places like our study village, it is a very common cultural practice to construct heaps of animal dung to use as compost when the next season comes. Further studies are needed to assess whether dung plays a direct role in transmission, including investigations into whether juvenile sandflies can emerge out of dung.

The independent risk factors in our study indicate exposure close to home (animal dung) and further away from the household (areas where hyraxes reside). Potentially, there could be different kinds of transmission cycles in villages such as our study village. First, transmission to humans could occur further away from home when spending time after dusk, when sandflies are most active [11], in areas where hyraxes reside. Infected sandflies could fly from these areas closer the individual's homes, leading to peri-domestic transmission. Finally, there are indications that *L aethiopica* can be transmitted from human to human and hence the presence of sandflies in or around the house could lead to anthroponotic transmission [11, 26]. We previously found sandflies (including some infected with *Leishmania*) present in and around houses in multiple settings, with their activity starting from the early evening and peaking by midnight [11]. Although not extensively studied, some findings suggest sand flies inside a house predominantly feed on humans [11].

A better understanding of the complex transmission cycle of CL in Ethiopia and when and where humans get infected is pivotal to designing effective control interventions. Preventive interventions to be explored could be targeted at the level of the household or the compound (clearance of animal dung, destruction of vector breeding places) [28]. Behavioural interventions such as educating the community about the use of insect repellents or wearing protective clothing when going outside at night or for certain activities such as collection of firewood could also be explored but might not be straightforward to implement. Another strategy that could be considered would entail eliminating hyraxes from areas that are densely inhabited by humans or frequently visited by humans.

Strengths of the study include the high response rate in the study and the comprehensive data collection on housing conditions, animal ownership, the peri-domestic environment as well as specific behaviour or features further from home. One key limitation is that the main outcome variable in our risk factor analysis was a combination of active CL (with relatively few cases) and CL scars (much more common). As CL scars occurred months to years before the survey, some risk factors that we captured might have changed over time (*e.g.* animal

ownership, behaviour). Entomological data would have allowed better insights into where transmission is occurring. Other limitations include the use of non-health workers to collect the CL suspected cases, the small number of PCR-confirmed CL cases, and the inclusion of only one village. In our study, we did not identify the causative species, although all previous surveys from the last ten years only identified *L aethiopica* from this area [22]. Information for some risk factors was based on observations during the survey but some factors such as the presence of dung can vary over time.

In conclusion, this is the first report on the existence of CL in Bilala Shaye, a rural village in the South of Ethiopia. While the prevalence of active CL was low, the high prevalence of CL scars indicates long-term endemicity. This calls for strengthening national surveillance systems. Besides increasing age, spending time outside in the evening (close to where hyraxes reside) and the presence of animal dung around the house were identified as independent risk factors for CL. In-depth studies should be done to further the understanding of the epidemiology and transmission of CL in affected areas, as a way to define interventions to prevent disease transmission.

## Supporting information

**S1 Table. House, environmental, and behavioral characteristics in the cutaneous leishmaniasis (CL) in the CL prevalence survey, Bilala Shay, Ethiopia 2021 (N = 252).**
(DOCX)

**S2 Table. Behavioral risk factors by age and sex in the cutaneous leishmaniasis prevalence survey, Bilala Shay, Ethiopia 2021 (N = 1012).**
(DOCX)

**S3 Table. Association between outdoor activities during late evening and spending time outside in areas where hyraxes reside in the CL prevalence survey, Bilala Shay, Ethiopia 2021 (N = 1012).**
(DOCX)

**S4 Table. Univariate association between socio-demographic and behavioural characteristics and cutaneous leishmaniasis (CL) in the CL prevalence survey, Bilala Shay, Ethiopia 2021 (N = 1012).**
(DOCX)

**S5 Table. Univariate association between household features and cutaneous leishmaniasis (CL) in the CL prevalence survey, Bilala Shaye, Ethiopia 2021 (N = 1012).**
(DOCX)

**S1 Data. Baseline data of the study population, Bilala Shaye; South Ethiopia 2021.**
(CSV)

## Acknowledgments

We are very grateful to all the study participants, who volunteered to participate in this study. We would like to thank the kebele administration of Bilala Shaye, the AMU-IUC project support unit, and the RUNRES-ETH project office for their support in the fieldwork. We also thank the AMU-HDSS field data collectors for enduring the hardship of data collection during the rainy seasons in Bilala Shaye. We thank the University of Gondar for providing the PCR tests.

## Author Contributions

**Conceptualization:** Behailu Merdekios, Mesfin Kote, Jean-Pierre Van Geertruyden, Johan van Griensven.

**Data curation:** Behailu Merdekios, Mesfin Kote, Myrthe Pareyn, Jean-Pierre Van Geertruyden, Johan van Griensven.

**Formal analysis:** Behailu Merdekios, Mesfin Kote, Myrthe Pareyn, Jean-Pierre Van Geertruyden, Johan van Griensven.

**Funding acquisition:** Jean-Pierre Van Geertruyden.

**Investigation:** Behailu Merdekios.

**Methodology:** Behailu Merdekios, Mesfin Kote, Myrthe Pareyn.

**Project administration:** Johan van Griensven.

**Resources:** Johan van Griensven.

**Software:** Johan van Griensven.

**Supervision:** Jean-Pierre Van Geertruyden, Johan van Griensven.

**Validation:** Behailu Merdekios, Myrthe Pareyn, Johan van Griensven.

**Visualization:** Behailu Merdekios.

**Writing – original draft:** Behailu Merdekios, Johan van Griensven.

**Writing – review & editing:** Behailu Merdekios, Mesfin Kote, Myrthe Pareyn, Jean-Pierre Van Geertruyden, Johan van Griensven.

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
