## [Decision Letter · Decision Letter 0]

28 Dec 2023

PONE-D-23-30018Prevalence and risk factors of cutaneous leishmaniasis (CL) in South-Ethiopia’s newest endemic site for CL.PLOS ONE

Dear Dr. Merdekios,

Thank you for submitting your manuscript to PLOS ONE. After careful consideration, we feel that it has merit but does not fully meet PLOS ONE’s publication criteria as it currently stands. Therefore, we invite you to submit a revised version of the manuscript that addresses the points raised during the review process.

We look forward to receiving your revised manuscript.

Kind regards,

A. K. M. Anisur Rahman, Ph.D.

Academic Editor

PLOS ONE

Journal Requirements:

3. In the online submission form, you indicated that your data will be submitted to a repository upon acceptance.  We strongly recommend all authors deposit their data before acceptance, as the process can be lengthy and hold up publication timelines. Please note that, though access restrictions are acceptable now, your entire minimal  dataset will need to be made freely accessible if your manuscript is accepted for publication. This policy applies to all data except where public deposition would breach compliance with the protocol approved by your research ethics board. If you are unable to adhere to our open data policy, please kindly revise your statement to explain your reasoning and we will seek the editor's input on an exemption. 

4. We note that Figures 1 and S1 in your submission contain map images which may be copyrighted. All PLOS content is published under the Creative Commons Attribution License (CC BY 4.0), which means that the manuscript, images, and Supporting Information files will be freely available online, and any third party is permitted to access, download, copy, distribute, and use these materials in any way, even commercially, with proper attribution. For these reasons, we cannot publish previously copyrighted maps or satellite images created using proprietary data, such as Google software (Google Maps, Street View, and Earth). For more information, see our copyright guidelines: http://journals.plos.org/plosone/s/licenses-and-copyright.

a. You may seek permission from the original copyright holder of Figures 1 and S1 to publish the content specifically under the CC BY 4.0 license.  

5. We note that Figure 1 in your submission contain copyrighted images. All PLOS content is published under the Creative Commons Attribution License (CC BY 4.0), which means that the manuscript, images, and Supporting Information files will be freely available online, and any third party is permitted to access, download, copy, distribute, and use these materials in any way, even commercially, with proper attribution. For more information, see our copyright guidelines: http://journals.plos.org/plosone/s/licenses-and-copyright.

Reviewers' comments:

Reviewer's Responses to Questions

**Comments to the Author**

1. Is the manuscript technically sound, and do the data support the conclusions?

Reviewer #1: Partly

Reviewer #2: Partly

Reviewer #3: No

2. Has the statistical analysis been performed appropriately and rigorously? 

Reviewer #1: No

Reviewer #2: No

Reviewer #3: No

3. Have the authors made all data underlying the findings in their manuscript fully available?

Reviewer #1: No

Reviewer #2: Yes

Reviewer #3: Yes

4. Is the manuscript presented in an intelligible fashion and written in standard English?

Reviewer #1: Yes

Reviewer #2: Yes

Reviewer #3: No

5. Review Comments to the Author

Reviewer #1: Abstract

Except the conclusion part, it is well written and explanatory.

Major correction

- The conclusion part of the abstract needs significant modification, which is not related to the objective, method, and result.

Minor correct

- Method part of the abstract

-'... features were collected using standardized questionnaires. Univaria...' == "standardized" ==> it is not indicated in the method section (of the study) about the method of standardization of the tools. Better to remove it.

Introduction

- Well written

- Minor correction=

- The citation has problems. For instance, the 2nd paragraph 1st statement. The indicated information is not available in the references they authors cited.

- On the 9th line (same paragraph), the start by describing 'Two studies' and putting three references....

The authors need to make this correction through the introduction part.

Method

Minor

Study design

- 'An exhaustive cross-sectional house-to-...'. The term 'exhaustive' is subjective. Remove it.

- '... Demography and Health Survey (18); activities ...' The reference and the information are not related.

Statistical analysis

- 'EpiData (version 2.2.3.187, EpiData Association, Odense, Denmark) was...': Which data was entered using 'EpiData'?

Results

Individual and household characteristics

- 'A total of 1012 individuals from 252 households were included with 100% response rate.': In the method part, it is indicated that the number of village residents were 1255. Here the total number of participants are 1012 and the response rate is 100%. How?

- Metal roof == Iron sheet

- Marital status==Caution: The proportion also included NA. (Example: 38% married =It does not mean those eligible for marriage)

- Table 2. The title: It is not only per age group, rename the table

- the use of 'patients' term. The study participants were not the patients. Proper use of terms

Discussion

- '... However, it is a very common cultural of Gamo people living in places like this to practice...': 'places like this' this need to be explain properly to remove ambigity

Major

Conduct of the survey

- '...farmland, gorges or areas where hyraxes reside, ...' The method to determine the reside of hyraxes needs to be started properly. Is it the presence of gorges or other method was used to determine the hyraxes reside? If yes, indicated it or if not better to use only 'presence of gorges' only.

- '...confirm CL by PCR in a sub-group, nine random individuals with active lesions compatible...' In this part, the authors should explain more. For instance, here are some missing information. Why only nine subjects only? Brief description about the random selection of the 9 cases. Who collected the sample? etc should have to be added.

Statistical analysis

- '... Factors with a P-value < 0.05 in univariate analysis were ...': In the result part (multiple regression part) and supplement tables, I did not see the reason for considering this cut-off point (for p-value). The number of IVs were not large in number (there are only 3 variables were included). So, why the p-value<0.05 for selecting the IVs. The adjustment needs inclusion of sound variables with correcting the cutoff point for p-value. In addition, all the included variables need to be described in the table.

Ethical issues

- What about those participants with active CL cases? Did the research provide treatment. If yes, what was the treatment? If no, why it was not?

Results

- The last citation of 1st paragraph of the result part is Table S1.: There is mismatch while citing tables. Be careful and check again to match the description with the tables

- '...located within 300 meters of areas where hyraxes ...': The method part did not explain properly the data collection including the 'hyraxes reside' and the method for estimating the distance

- Table 3. The description in the method section and the table are not consistent. The authors described that the p<0.05 in bivariant analysis were included in the multivariate. However, they did not include those variables. This need to be corrected or explained in the method part properly.

- In the sensitivity analysis ... : The authors need to explain the reason for doing sensitivity analysis for those less than 12 years in the method part.

-

Discussion

- Our findings call for national evaluations in CL risk areas to more exactly define the current epidemiology and the need of care in the country: The argument used in this and previous studies does not lead for this type of conclusion. Because (1) This study is in only one village (2) The previous studies still did not show any difference with the national mapping (3) The authors did not indicate any finding different with the national evidence currently used by the MoH of Ethiopia. The study area is also indicated as the high endemic area for CL on the national mapping. So, what are the base for this type of description?

- '...previously detected Leishmania parasites at the nose of one single goat (28)’: It is well explained regarding having goats at home as a risk factor. However, the authors also include the goat rearing situation in Ethiopia. It is obvious that those who are living near the gorges area commonly have goats. That might support the evidence and also give direction for future research to check the pattern more carefully.

- The 8th paragraph of the discussion part: 'A better understanding of the complex .... abundancy of infectious sand flies.': What is the importance of this paragraph for explaining this finding. It is not related to the study, and it doesn’t providing explanation based on the study ==> better to remove the whole paragraph

- Include the following information at the strength and limitation part of the discussion

o The study depends only in one village based on the disease distribution.

o The data collectors: Physical examination was done by non-health workers

o The use of sub-sample for confirmatory examination of the samples using PCR

- The conclusion of the study: It is not related to the objective, method followed and finding of the study. Better to consider the finding and limitation (including the study is based on one village)

Reviewer #2: Review on “Prevalence and risk factors of cutaneous leishmaniasis (CL) in South-

Ethiopia’s newest endemic site for CL”

This is an important manuscript that described the findings of a community survey on prevalence and risk factors of cutaneous leishmaniasis (CL) in a village in southern Ethiopia where CL was previously not reported.

Authors may consider the following comments to improve the manuscript:

General

As discussed by the authors in the third paragraph of the discussion, the presence of high proportion of CL related scars indicates that the area is already endemic for CL. The area might not have been recognized as endemic due to poor surveillance and reporting system. Thus, this should not be reported as “newest” endemic site. Rather, it is important to emphasize the fact that poor surveillance and reporting system is missing to recognize such endemic sites with ongoing transmission, and thus, leaving behind such public health issues unaddressed.

On the other hand, may the low number of active CL than CL scar be a sign that CL is actually decreasing in the area?

Some of the independent variables considered in the risk factor analysis are not independent to each other. For example, housing conditions may be associated with income, a house with better roofing will also have better wall, outside kitchen and toilet. Similarly, the presence of animal dung in the compound will be related to keeping animals in the compound or the house. Spending outside where hyraxes reside, outside activities and spending time outside in the evening are overlapping. Analyzing such related and overlapping variables in the same model looking for associations may lead to multicollinearity bias. Authors need to clearly explain how such possible bias was mitigated.

Specific comments:

Title and short title: better to avoid using the terms “newest” and “new” as explained above

Authors summary: The statement “The study highlights the importance of effective vector control and prevention strategies that focus on limiting human-sandfly contact in areas with hyrax presence” is generally correct statement but is not in the scope of the study findings.

Introduction: The references mentioned in the statement, “There are however several areas in southwestern Ethiopia with similar environmental and geographical features from which CL is not reported (9,15).” to justify the absence CL case from the study area and present it as a new endemic area, do not describe the absence of CL in the study area in the past, they are CL reports from other endemic areas.

Methods:

- It is described that 9 out of the 25 individuals with active lesions compatible with CL were selected randomly for PCR testing of whom eight were confirmed. It is important to describe how the clinical diagnosis of CL and CL scar was made and distinguished from other similarly manifesting skin conditions.

- The purpose of the sensitivity analysis needs to be explained.

Results:

- High proportion of households (86%) have indoor smoke production. This finding is not discussed in terms of its potential as insect repellent and protective means from sandfly. May this be a cultural means developed by the community to chase away different insects? May the low number of active CL as compared to the high CL scar without interventions in the area indicate that the disease is decreasing with some cultural interventions like this? On the other hand, it is also important to indicate this major side finding as a potential respiratory disease risk in the community (although not the objective of the study).

- It is strange that there are no cows and oxen owned by the study population that account 24% farmers. Oxen are common farming animals in Ethiopia.

- Eucalyptus is a common tree planted around household in Ethiopia and used for different purposes. It is thought to have some insect repellent effect. Please describe if such data was collected.

- Table 2 shows 13 active CL and 50 CL scar in under 12 years. But in the last paragraph of the results section talking about sensitivity analysis the number is shown as 80. Please harmonize.

- S2 Table: Analysis for reasons for activities outside shows 49.72% (for CL) vs 50.3% (for n CL) for fetching water/firewood with OR of 2.3 (1.6 – 3.6). It is good to revisit the analysis as such almost equal numbers may not show such statistical difference. The percentage of most of the other variables also favors no CL than CL as is discussed.

- Fig: S1: Indicating where the gorges and the habitats of hyrax are on the map will be more informative.

Discussion

- For the studies referenced (#25, #14) in the second paragraph of the discussion, clearly state if these studies have not included the current study site.

- For the statement, “Several CL outbreaks have been identified over the last years, all incrementing L aethiopica.” please included references.

- Please consider the comments in the other sections in the discussion.

Reviewer #3: 1. Editorial

1.1 What does “newest endemic site” or “A new CL hotspot” mean? Does it mean the study area was not endemic for CL previously. If so, what is the evidence for that? The words “new” and “newest” are misnomers, and they should be replaced from the title and from the main body of the manuscript.

2. Last sentence of the “Introduction” in page 11 begins by the phrase “Our findings aim to….”.

It makes sense to change it to “Our research (This research) aims to contribute to ……”

2. Regressions

2.1 Combining active CL and past CL lesions in defining a key outcome variable

Risk of CL was a key outcome variable assessed by combining active and past CL cases for the purpose of uni-(multi-) variate analysis using regression models. There were 382 cases with past lesions (390 according to table 2) whose median age was 32. Assuming most active lesions occur during childhood, the majority of the 382 cases with scars may have occurred 20 years ago. Clearly, active and past CL lesions were temporally distant from each other. Consequently, combining the active lesions with scars in formulating the outcome variable cannot be straightforward particularly that data is based on a proportionally larger of past CL lesions (scars).

For instance, S2 Table gives the univariate analysis with a breakdown by age, and shows that not only children 5 to 17, but also adults have a higher odds of CL risk (OR = 6.6 – 9.4). However, from Table 2, it was clear that children in the age bracket 5 to 11 had the highest prevalence of active CL (7.6%), and those under 5 had 1.5% prevalence, and children older than 11 years and adults had low prevalence rates (ranging from 1.0% to 2.0%). Therefore, it was clear that combining active and healed lesions was a mistake.

2.2 Domestic animals and animal dung

Even though all households had animals, animal dung was found in 77% of households. One wonders why all household did not have dung in the compounds. Furthermore, it seems plausible to assume that presence of animals and animal dung are not independent of each other. In such a scenario, the regression model cannot assume these two variables to be independent of each other.

In the univariate analysis, the presence of goats/sheep inside houses (but not in the compound) seemed to show marginal association with the outcome variable. Given the flaws in the design/analysis, caution should be made not to over-interpret the findings. The authors tried to build on this observation as an important finding (see discussion section implying attraction of sandflies by goats/sheep; also referring to a single natural infection of a goat - citing a publication of their own).

In the multiple regression section, it is interesting to note that presence of domestic animals in the households did not come out as a predictor of CL incidence whereas animal dung turned out to be a remarkably significant predictive variable (univariate analysis, S3 Table).

2.3 The steps involved in generating Table 3 and the adjustments made variables-wise is not clearly presented. The structure of the table is not self-explanatory. A detailed description of the model is lacking.

3. Table 1

For the section of marital status, does it make sense to re-calculate the percentages by taking the denominator as 767 (those >18 yrs)? This gives 36.5%, 49.8% and 13.7% marital status for single, married and widowed/separated respectively.

4. Strengths and limitations

4.1 Why would the use of molecular techniques be considered as a strength? By the same token, the total population sampling seems to be related to feasibility rather than scientific justification. Apparently, the total population of the study population is small which must have incentivized the investigators to do it all without sampling. Of course, the approach minimizes sampling error.

4.2 The limitations have been pointed out appropriately but remains to be a concern. As explained above, there is a deduction that older children and adults are at higher risk of acquiring CL. A close look at the data seems to point that in fact small children in the age bracket 5-11 are the ones at highest risk.

4.3 Because the key outcome variable was comprised of both active and healed lesions, interpretation of associations between the several presumed explanatory variables and the key outcome variable remains to be obscure, and the conclusions made are not fully substantiated.

5. References

5.1 Ref # 16 does not give list of authors.

5.2 Journal names are given either abbreviated or written in full. Consistency is advised.

6. Others

6.1 Editorial corrections will improve the quality of the manuscript.

6. PLOS authors have the option to publish the peer review history of their article (what does this mean?). If published, this will include your full peer review and any attached files.

Reviewer #1: **Yes: **Befikadu Tariku Gutema

Reviewer #2: No

Reviewer #3: No

---

## [Author Response · Author response to Decision Letter 0]

3 Aug 2024

Journal Requirement:

We believe that the manuscript has followed the submission guidelines and styles for the journal.

Regarding the data availability, We have uploaded the excel sheet that holds at least some data for your further look, if you need it (Supporting information)

Question 4 & 5: The Figures 1 and S1 in my submission contain map. This map is created using my coordinate points on the ground and created for my specific study. It is my work and confirm that there could not be any external body to claim for the copyright. 

Reviewers' comments:

Please refer to the separate file for it as "Response to Reviewers Comment" in the attached files in the resubmission.

---

## [Decision Letter · Decision Letter 1]

21 Aug 2024

PONE-D-23-30018R1Prevalence and risk factors of cutaneous leishmaniasis in a newly identified endemic site in South-EthiopiaPLOS ONE

Dear Dr. Merdekios,

Thank you for submitting your manuscript to PLOS ONE. After careful consideration, we feel that it has merit but does not fully meet PLOS ONE’s publication criteria as it currently stands. Therefore, we invite you to submit a revised version of the manuscript that addresses the points raised during the review process.

**ACADEMIC EDITOR: **Please carefully review the comments and suggestions made by the reviewers, make the necessary adjustments to the manuscript, and then resubmit it.

We look forward to receiving your revised manuscript.

Kind regards,

Rajib Chowdhury, M.Sc.; MPH

Academic Editor

PLOS ONE

Journal Requirements:

Additional Editor Comments:

Please carefully review the comments and suggestions made by the reviewers, make the necessary adjustments to the manuscript, and then resubmit it.

Reviewers' comments:

Reviewer's Responses to Questions

**Comments to the Author**

1. If the authors have adequately addressed your comments raised in a previous round of review and you feel that this manuscript is now acceptable for publication, you may indicate that here to bypass the “Comments to the Author” section, enter your conflict of interest statement in the “Confidential to Editor” section, and submit your "Accept" recommendation.

Reviewer #1: All comments have been addressed

Reviewer #3: (No Response)

Reviewer #4: All comments have been addressed

2. Is the manuscript technically sound, and do the data support the conclusions?

Reviewer #1: Yes

Reviewer #3: Partly

Reviewer #4: Partly

3. Has the statistical analysis been performed appropriately and rigorously? 

Reviewer #1: Yes

Reviewer #3: I Don't Know

Reviewer #4: Yes

4. Have the authors made all data underlying the findings in their manuscript fully available?

Reviewer #1: Yes

Reviewer #3: Yes

Reviewer #4: No

5. Is the manuscript presented in an intelligible fashion and written in standard English?

Reviewer #1: Yes

Reviewer #3: Yes

Reviewer #4: Yes

6. Review Comments to the Author

Reviewer #1: Data collection and entry

Line 121: ‘… designed and uploaded to a tablet with Open Data Kit software for data collection.’

Line 159: ‘EpiData (version 2.2.3.187, EpiData Association, Odense, Denmark) was used for data entry…’

You indicated that the tools were uploaded on ODK software. Again, you also indicated the data was entered on ‘EpiData’. In your response you mentioned that ‘All data mentioned under ‘conduct of the survey’ were entered using EpiData.’ If that is the case, you have to remove that statement indicating ODK. Or, indicate which part of the data was collected using paper and ODK.

Line 230: '... of 2.5% (5% confidence interval ...' Make the correction 5% =95%

Reviewer #3: In Line 281 of the manuscript, there is the following statement "Besides increasing age, spending time outside in the evening (close to where hyraxes reside) and the presence of animal dung around the house were identified as independent risk factors for CL" that hinges to the point that increasing age is a risk for CL. I have tried to point out that lumping the active and healed lesions data will be leading to scientific misinformation and misdirection of control efforts.

The authors have recognized this as a limitation and have toned down this issue overall, but retained the phrase " Besides increasing age...".

I suggest the phrase "Besides increasing age" be removed from the statement above.

Reviewer #4: Dear editor and research team,

Thank you for sharing this work, and as advance disclaimer, I was not part of the first review round of the manuscript - so I am thankful to have the chance to look at the revision, including the comments from 3 reviewers and the answers.

The key concerns I had when reading the original manuscript, were mostly queried (namely: the far reaching conclusion that may be too far fetched relevant to the finding, choice of methodology of cross sectional household survey, and also on statistical analysis approaches and finding).

Few minor comments from my side that I think not yet addressed satisfactorily in the revision:

- The conclusion in the abstract, which highlight the confirmed endemicity of this one village still does not 'reflect' the benefit of this research to the community surveyed: is the village gonna be included in the national surveillance.. or there will be access to diagnosis and treatment?

- Related to that, the choice of this village to be exhaustively surveyed is still not clear for me- inR1 page 6 line 103- refer to wider survey of 38 villages, and the only justification to choose this is the steep valley separating it from Ochollo and also the small size of residents? Was it the case? What motivated the team to choose this one, if only just feasibility?

- Regarding combining CL lesion and scars- this seems indeed quite common, as in this reference (https://journals.plos.org/plosntds/article?id=10.1371/journal.pntd.0006914).

- The method stated the survey took place in 2021 but some tables refer to 2020, e.g Table 3, Table S2-S5 - which one is correct?

Other than that, the authors have shown serious attempts to revise the paper and improve it based on the extensive comments they get.

7. PLOS authors have the option to publish the peer review history of their article (what does this mean?). If published, this will include your full peer review and any attached files.

Reviewer #1: **Yes: **Befikadu Tariku Gutema

Reviewer #3: No

Reviewer #4: No

---

## [Author Response · Author response to Decision Letter 1]

24 Sep 2024

We have made the minor correction edition, supported by response to reviewers' comment and now it is in its full clean version ready to take it to the next level.

Hope this would be the final.

I thank you.

---

## [Editor Report · Decision Letter 2]

27 Sep 2024

Prevalence and risk factors of cutaneous leishmaniasis in a newly identified endemic site in South-Ethiopia

PONE-D-23-30018R2

Dear Dr. Merdekios,

We’re pleased to inform you that your manuscript has been judged scientifically suitable for publication and will be formally accepted for publication once it meets all outstanding technical requirements.

Kind regards,

Rajib Chowdhury, M.Sc.; MPH

Academic Editor

PLOS ONE
---

## [Editor Report · Acceptance letter]

18 Oct 2024

PONE-D-23-30018R2 

PLOS ONE

Dear Dr. Merdekios, 

I'm pleased to inform you that your manuscript has been deemed suitable for publication in PLOS ONE. Congratulations! Your manuscript is now being handed over to our production team.

Kind regards, 

on behalf of

Dr. Rajib Chowdhury 

Academic Editor

PLOS ONE